# Exogenous Application of Melatonin to Green Horn Pepper Fruit Reduces Chilling Injury during Postharvest Cold Storage by Regulating Enzymatic Activities in the Antioxidant System

**DOI:** 10.3390/plants11182367

**Published:** 2022-09-11

**Authors:** Luyao Wang, Xuemeng Shen, Xiumei Chen, Qiuli Ouyang, Xiaoli Tan, Nengguo Tao

**Affiliations:** School of Chemical Engineering, Xiangtan University, Xiangtan 411105, China

**Keywords:** green horn pepper fruit, melatonin, chilling injury, reactive oxygen species, antioxidant system

## Abstract

Chilling injury (CI) caused by exposure to low temperatures is a serious problem in the postharvest cold storage of pepper fruit. Melatonin (MT) has been reported to minimize CI in several plants. To evaluate the effectiveness of MT to minimize CI in green horn pepper and the possible mechanism involved, freshly picked green horn peppers were treated with MT solution at 100 μmol L^−1^ or water and then stored at 4 °C for 25 d. Results showed that MT treatment reduced CI in green horn pepper fruit, as evidenced by lower CI rate and CI index. MT treatment maintained lower postharvest metabolism rate and higher fruit quality of green horn peppers, as shown by reduced weight loss and respiratory rate, maintened fruit firmness and higher contents of chlorophyll, total phenols, flavonoids, total soluble solids and ATP. Additionally, the contents of hydrogen peroxide, superoxide radical, and malondialdehyde were kept low in the MT-treated fruit, and the activities of the enzymes peroxidase, superoxide dismutase, and catalase were significantly elevated. Similarly, the ascorbate–glutathione cycle was enhanced by elevating the activities of ascorbate peroxidase, glutathione reductase, dehydroascorbate reductase, and monodehydroascorbate reductase, to increase the regeneration of ascorbic acid and glutathione. Our results show that MT treatment protected green horn pepper fruit from CI and maintained high fruit quality during cold storage by triggering the antioxidant system

## 1. Introduction

Horn pepper is one of the main green pepper varieties in southwest China, with a distinct spicy taste, dark green peel, and is quite rich in vitamin C, flavonoids and total phenols [1,2]. However, the pepper fruit has a short postharvest lifespan [3]. Low-temperature storage and improved transportation technology are widely used to extend the storage life [4,5,6]. However, peppers are sensitive to low temperatures and susceptible to chilling injury (CI), which is characterized by softening, shriveling, tissue depression, and the formation of water-soaking spots, a predisposition to decaying [7,8,9].

In plants, several hypotheses have been proposed to explain the occurrence of CI, among these, imbalance in reactive oxygen species (ROS) metabolic homeostasis is considered the primary cause [10,11]. Plants have antioxidant enzymes that directly eliminate specific ROS and non-enzymatic antioxidants (such as ascorbic acid (AsA), glutathione (GSH), phenols and flavonoids) that indirectly scavenge the ROS. Under normal physiological conditions, the plants can maintain a homeostatic balance between ROS production and ROS scavenging [12,13]. However, certain unfavorable factors, such as prolonged low temperature and impaired cellular ROS-scavenging systems can lead to excessive accumulation of ROS, thereby disrupting ROS homeostasis [14]. Sustained increases in ROS and malondialdehyde (MDA) levels have thus been observed in fruit such as bananas [15], sweet cherries [16], blood orange [17], and sapota [18] when stored at low temperatures. The resultant oxidative stress can be cytotoxic, leading to CI. Therefore, strict control of intracellular ROS homeostasis is crucial for the maintenance of normal cellular metabolism and to reduce the occurrence of CI.

Melatonin (N-acetyl-5-methoxy-tryptamine, MT) is a multifunctional bio-signaling molecule that is widely distributed in many organisms, and has been proven to be beneficial to human health [19,20,21]. Due to its convenient use, obvious effect on the preservation of fruit and vegetables, and can be used as a food supplement, the application of MT in the postharvest industry has broad potential [22,23]. As a broad-spectrum potent free radical scavenger and antioxidant, MT is involved in plant response to various stresses including CI [24,25,26]. In kiwifruit, MT alleviated CI during cold storage by promoting the accumulation of non-enzymatic antioxidants and increasing the activity of enzyme antioxidants [27]. In green bell pepper fruit, MT treatment reduced fruit CI by regulating membrane lipid metabolism and ROS accumulation [28]. And in guava [29], button mushroom [30] and blueberry fruit [31], MT not only reduced CI, but also maintained a high nutritional quality during cold storage. These studies suggest that MT is effective at controlling CI and may be applied in the pepper industry to reduce the associated postharvest losses. However, the ability of MT to control the fruit quality and CI of green horn peppers has not been studied. Whether exogenous MT treatment can enhance the cold resistance of postharvest horn pepper by regulating the antioxidant system needs further evidence.

In this study, the effects of 100 μmol L^−1^ of MT on the CI and fruit quality of green horn pepper fruit stored at low temperature were investigated. We systematically examined the effects of MT treatment on antioxidant enzymes and the AsA–GSH cycle. This study aimed to provide a biochemical and physilogical basis for the commercial use of MT to reduce CI and quality loss in green horn peppers during postharvest cold storage.

## 2. Results

### 2.1. MT Treatment Alleviated CI in Postharvest Horn Peppers

Changes in the surface appearance of pepper fruit during cold storage with or without MT treatment are shown in Figure 1. After 15 days of storage at 4 ℃, CI symptoms such as water spots, tissue depressions, and reduction in surface glossiness appeared on the surface of fruit in the control group (Figure 1A). MT treatment delayed the development of CI symptoms to day 20 as evidenced by lower CI rate (Figure 1B) and CI index (Figure 1C), and a normal surface glossiness. On 25 d, the CI rate and CI index of fruit in the MT treated group were only 28% and 20%, respectively, of those in the control group.

### 2.2. MT Treatment Reduced Postharvest Metabolic Rate and Quality Loss of Horn Pepper Fruit

Within the first 5 days of cold storage, the rate of weight loss in fruit within the control and MT-treated groups were the same. Beyond 5 d, MT-treatment significantly lowered weight loss in horn pepper fruit such that by 25 d, it was 29% lower than the control samples (Figure 2A). As weight loss increased, fruit firmness decreased, resulting in fruit softening and a decline in fruit quality. However, MT treatment maintained a higher firmness of pepper fruit during the entire storage period (Figure 2B). Generally, respiration rate and ATP content also declined during cold storage. However, MT-treatment greatly lowered the rate of decline in respiration rate such that by 15 d, it was only 62.20% of that in the control group (Figure 2C). MT treatment also lowered the rate of decline in ATP content such that by 25 d, MT treated fruit had 27% more ATP than fruit in the control group (Figure 2D).

Chlorophyll content gradually and steadily decreased in the control samples during the entire cold storage period. MT treatment significantly lowered the decline in chlorophyll content such that by 25 d, the chlorophyll content was 24% higher than in the control group (Figure 2E). The TSS content of pepper fruit in both groups increased until 15 d of storage and thereafter declined. However, MT treatment maintained a significantly higher TSS content from 10 d of incubation. By 15 d of storage, the TSS content of MT-treated fruit was 14% higher than in the control group (Figure 2F). Generally, the contents of total phenols and flavonoids initially increased during the cold storage period and later declined. MT treatment maintained higher total phenolic and flavonoid contents throughout the storage period such that by 15 d, the total phenolic and flavonoid contents in the MT treatment group were 42% and 19% higher than those in the respective control groups (Figure 2G,H).

### 2.3. MT Treatment Alleviated ROS Accumulation and Lipid Damage

There was a general increase in the contents of H_2_O_2_ and O_2_^•−^ in both the control and MT-treated fruit during the cold storage period. However, MT-treatment resisted increase in the contents of both H_2_O_2_ and O_2_^•−^ such that by 25 d, the values in the control group were 1.59 and 1.07 times higher than in the respective MT treatment groups (Figure 3A,B). The increased ROS concentrations led to peroxidation of membrane lipids, as evidenced by the steady increase in MDA content. MT treatment however, alleviated membrane lipid damage by reducing ROS accumulation and thus maintaining lower MDA contents throughout the storage period (Figure 3C).

### 2.4. MT Treatment Increased the Contents of AsA and GSH

Higher levels of AsA and GSH can maintain a higher reduction potential within cells, thereby improving cellular antioxidant capacity [13]. In our study, the AsA content in the fruit within the control group decreased all through the storage period. MT treatment maintained a nearly normal level of AsA such that by 25 d, the AsA content of the treatment group was still 60% higher than that in the control (Figure 4A). AsA is usually oxidized to form DHA. As AsA content steadily decreased in the control samples, higher levels of DHA were recorded. MT treatment however, maintained lower levels of DHA throughout the treatment period, signifying reduced oxidation of AsA (Figure 4B). A high AsA/DHA ratio was thus maintained in the MT-treated group, especially during late storage period (Figure 4C). Whereas there was a general decrease in the content of GSH during the cold storage period, it was not a smooth decline as sometimes it increased or decreased (Figure 4D). MT treatment maintained a relatively higher GSH content than in the control samples. GSSG content steadily increased throughout the 25 d of storage, with MT treatment maintaining significantly lower levels of GSSG (Figure 4E). Overall, the ratio GSH/GSS declined during the cold storage period, but MT treatment maintained relatively higher GSH/GSSG ratio within the horn pepper fruit (Figure 4F).

### 2.5. MT Treatment Enhanced the Activities of Antioxidant Enzymes

The APX activity in pepper fruit within the control samples slightly increased within the initial 15 d then decreased thereafter. MT treatment triggered a rapid increase in APX activity to reach a peak value on 20 d and then decreased to a value significantly higher than in the control samples by 25 d (Figure 5A). There was a general increase in the activity of GR during the cold storage period. MT-treated fruit maintained relatively high GR activity from 5 d of storage (Figure 5B). DHAR activity decreased during the storage period. However, MT treatment resisted decrease in the DHAR activity such that the recorded values at any given time were higher than in the control samples (Figure 5C). The MDHAR activity of fruit in the control group remained approximately constant during the storage time. Treatment of fruit with MT activated a rapid increase in MDAHR activity to reach a peak on 5 d. Thereafter, a gradual decrease in MDAHR was recorded but the activity remained significantly higher than in the control group all through the storage period. By 20 d, the MDAHR activity was 67% higher in the MT-treated group than in the control (Figure 5D). The activities of POD, CAT and SOD increased for the first 15 d then decreased (Figure 5E–G). MT treatment induced higher activities of these antioxidant enzyme such that on 15 d of storage, POD, CAT and SOD activities were 99%, 48% and 112% higher than in the respective control groups.

### 2.6. Correlation Analysis of ROS Metabolism with Pepper CI and Fruit Quality

Sixteen of the indicators related to ROS metabolism were used to analyze the effect of low temperature on ROS metabolism of pepper fruit. The PLS-DA permutation test in 12 component and permutated 200 times showed that both R2Y and Q2 values were lower than the original points (Figure 6A), demonstrating that the PLS-DA model could be used for subsequent analysis. The hierarchical cluster analysis (HCA) plot and score scatter plot of PLS-DA displayed good reproducibility for each sample and apparent separation between MT treatment and control at 10, 15 and 20 d (Figure 6B,C). This indicated that the effect of MT treatment on pepper fruit was more pronounced from 10 d to 20 d. OPLS-DA was used to analysis the key ROS metabolism related indicators in MT-treated peppers, 6 characteristics were considered as the key antioxidant system related characteristics (VIP > 1), including DHA, GSH, APX, MDHAR, GR, POD and flavonoids (Figure 6D). Further analysis of the correlation between fruit quality and ROS metabolism suggested that during the cold storage, ROS accumulation was significantly positively correlated with CI rate and CI index of pepper, and significantly negatively correlated with the respiration rate, chlorophyll content and fruit firmness. The enhancement of antioxidant system was conducive for fruit quality maintenance (Figure 6E).

## 3. Discussion

### 3.1. MT Treatment Relieved the CI and Quality Loss of Horn Pepper Fruit

Green pepper fruit are sensitive to low temperature and are prone to CI under low temperature stress, which leads to a decline in fruit quality and market value. As a type of small indole compound widely present among different organisms, MT can improve the fruit resistance to various stresses [32]. For example, MT applications enhanced the chilling tolerance and reduced occurrence of CI in mango [33], cucumber [34] and kiwifruit [27]. In the present study, application of MT distinctly lowered the incidence of CI in horn pepper fruit. The treated fruit were preserved with a superior appearance and exhibited lower CI rate and CI index, which was consistent with the results of Kong et al. [28] on green bell peppers.

Active metabolic processes such as transpiration and respiration of postharvest fruit and vegetables are important factors leading to the senescence and quality loss of products [13,35]. Firmness is a crucial index reflecting the textural characteristics of fruit thus any changes in firmness directly affect the postharvest metabolic intensity and longevity [36]. In our present study, weight loss rate of pepper fruit increased during the cold storage, but MT treatment diminished the increment of fruit weight loss rate after 5 d. This may be attributed to the fact that MT treatment maintained lower respiration intensity and higher firmness of the pepper fruit, thereby reducing fruit water loss and metabolite consumption during storage. Additionally, the higher ATP content indicated that the pepper fruit in the treatment group retained sufficient energy for various metabolic processes. Energy metabolism is strongly associated with the occurrence of postharvest CI in crops [37]. Higher availability of energy is beneficial for the synthesis of cell membrane lipids and wound repair, which together retard membrane lipid peroxidation [38].

The occurrence of CI and active postharvest metabolism can lead to deterioration of fruit quality [39]. Changes in chlorophyll levels are an important sensory indicator for assessing the freshness of green products. Phenols and flavonoids are important non-enzymatic antioxidant components in pepper fruit, which are strongly associated with fruit resistance to coldness [40]. They also contribute to the removal of ROS and inhibition of membrane lipid peroxidation which together reduce the occurrence of CI [40]. Reducing the loss of these substances is conducive to the long term preservation of product quality [39]. Thus in the current study, pepper fruit treated with MT retained higher fruit quality and lower CI. Additionally, high levels of nutrients are key to sustaining cellular osmotic pressure and antioxidant capacity, and enhancing fruit tolerance to coldness. Similar regulation of the postharvest metabolism and quality by MT has been documented in broccoli [41] and apple fruit [42]. These findings illustrate that MT displays a promising potential to maintain the nutritional and commercial value of horn pepper fruit during cold storage.

### 3.2. MT Treatment Reduced Chilling Injury and Quality Loss by Enhancing the Antioxidant System in Horn Pepper Fruit

Sustained low temperature can cause disruption of cellular ROS metabolism, resulting in a sequence of oxidative damage that may lead to CI [43]. In this study, H_2_O_2_ and O_2_^•−^ continuously accumulate in the control pepper fruit, aggravating membrane lipid peroxidation as manifested by the continuous increase in MDA levels. This indicate that ROS metabolic homeostasis was disturbed by the chilling temperatures, which led to the development of pepper fruit CI in the middle to late stages of storage. Heat map correlation analysis further demonstrated that H_2_O_2_, O_2_^•−^ and MDA accumulation was positively correlated with the occurrence of fruit CI, and negatively correlated with weight loss rate, respiration rate and fruit firmness. These results indicate that imbalance in ROS metabolism was the main reason for the occurrence of pepper fruit CI. They also imply that postharvest metabolism accelerated the accumulation of ROS, leading to pepper fruit CI. However, MT treatment reduced the accumulation of H_2_O_2_, O_2_^•−^ and MDA throughout the storage period, effectively reducing oxidative damage to cells, thus alleviated the fruit from CI. Similar outcomes were previously reported in banana [15] and peache [44] fruit treated with MT during low temperature storage.

Activation of enzymatic and non-enzymatic antioxidant systems by MT is a key mechanism to diminish cellular ROS accumulation [14,45]. In the current study, the ability of MT treatment to maintain low contents of H_2_O_2_ and O_2_^•−^ in horn peppers may be attributed to the enhanced activities of POD, CAT and SOD, thereby reducing ROS accumulation and damage to cells. AsA and GSH are also critical antioxidants and their ratios of reduced to oxidized state (AsA/DHA and GSH/GSSG) can be indicative of the cellular redox status and antioxidant potential [46]. Generally, a high proportion of their reduced states will enhance the plant’s resistance to stress. Li et al. [47] observed that cold acclimation in tomato fruit involved stimulation of AsA and GSH accumulation, and increasing the ratios of AsA/DHA and GSH/GSSG. In our present study, MT-treated peppers maintained higher AsA/DHA and GSH/GSSG ratios and generally, a high efficiency of AsA-GSH cycling, hence improving the antioxidant capacity of the pepper fruit. APX, MDHAR, DHAR, and GR serve as critical catalysts involved in the AsA–GSH cycle. MT application improved the activities of these enzymes and promoted the regeneration and accumulation of AsA and GSH in the AsA–GSH cycle. This is consistent with the findings by Song et al. [46] in cryopreserved peach fruit.

To understand the role of antioxidant systems in MT treatment, we conducted PLS-DA analysis on 16 relevant indicators. Our results showed that the MT-treated samples were significantly dispersed from the control samples between 10–20 d, indicating that MT significantly regulated the antioxidant system of the stored fruit within this duration. This explains why MT treatment maintained higher fruit quality within the same period of time. MT is thus key to reducing CI on fruit and maintaining quality in horn pepper fruit during cold storage. The OPLS-DA analysis further clarified that the indicators of the antioxidant system including DHA, GSH, APX, MDHAR, GR, POD and flavonoids were the key indicators in this system and have important contributions to the role of MT, but the associated molecular mechanisms still need to be investigated. Nevertheless, our study has revealed that MT treatment boosted the activity of the antioxidant system in chilled horn pepper fruit and limited the accumulation of ROS and the associated cellular damage. These changes helped minimize the possibility of CI to pepper fruit and maintained a high fruit quality during cold storage.

## 4. Materials and Methods

### 4.1. Selection and Treatment of Fruit

Horn pepper fruit (Ciban) were harvested from a nearby greenhouse in Xiangtan province, China. Dark green pepper fruit of uniform size and no damage were used for the treatment. After washing and drying, the fruit were divided into two equal groups; the control group (treated with water), and the treatment group (treated with 100 µmol L^−1^ MT solution). All fruit were then air dried, put in a basket and sealed with a polyethylene bag, then incubated at 4 °C for 25 d. Fruit were monitored and sampled at 5 d intervals.

### 4.2. Measuremet of CI Rate and CI Index

The CI rate (%) and CI index were determined according to the method described by Yao et al. [7]. CI was evaluated based on the appearance of pitting symptoms on the fruit surface. Pepper fruit were also allocated to 5 classes depending on the percentage proportion of CI on the entire fruit surface; 0, no CI appeared (excellent quality); 1, <25% surface area appeared CI; 2, 25–50% surface area appeared CI; 3, 50–75% surface area appeared CI; and 4, >75% surface area appeared CI.

### 4.3. Determination of O_2_^•−^, H_2_O_2_ and MDA Contents

Estimation of O_2_^•−^ and H_2_O_2_ contents was performed according to the method described by Ding et al. [48]. The reaction between O_2_^•−^ and hydroxylamine produces NO_2_^−^, which is capable of reacting with aminobenzenesulfonic acid and α-naphthylamine to produce pink azo dyes with a maximum absorption wavelength of 540 nm. H_2_O_2_ reacts with titanium salts to form a yellow precipitate of peroxide-titanium complex. The precipitate can dissolve in concentrated sulfuric acid to form a solution with a maximum absorption peak at a wavelength of 415 nm. The content of O_2_^•−^ and H_2_O_2_ were measured in mmol kg^−1^.

MDA was detected by the thiobarbituric acid assay [43]. 1.0 g of fruit samples were weighed, crushed in trichloroacetic acid solution and centrifuged at 4 °C, 7000× *g* for 25 min. The supernatant was mixed with thiobarbituric acid and boiled in a boiling water bath for 30 min. The absorbance of the resultant solutions were measured and MDA content was calculated according to the method described by Zha et al. [49].

### 4.4. Weight Loss, Firmness, Respiration Rate and ATP Content

The weight loss rate was assessed according to the method by Tan et al. [35]. Pepper firmness was evaluated with a hand-held hardness tester (Gy-1, Yangzhou Huazhong Wujinjiaodian Chemical Co., Ltd., Yangzhou, China) and the results were expressed in Newtons (N). The respiration rate was monitored and estimated as previously described [50]. The rate of CO_2_ release from five pepper fruit were then recorded in triplicates over a period of 30 min. The ATP content was detected by phosphomolybdic acid colorimetry utilizing a commercial kit acquired from Resshineen Biotechnology Co., Ltd. (Guangzhou, China).

### 4.5. Fruit Quality

To measure the total soluble solids (TSS) content, pepper fruit tissue (5.0 g) was ground in a clean mortar, and a drop of the filtrate was placed on the prism of a portable refractometer to determine the refractive index [51]. The chlorophyll content was assessed according to the method by Tan et al. [52]. Briefly, 5.0 g of pepper fruit tissue from each of the treatment and control samples were ground in 20 mL of 80% acetone solution. The absorbance of the filtrate was then measured at 663 and 645 nm. Total phenol and flavonoid contents were extracted and assessed in accordance with the methods of Duan et al. [53].

### 4.6. Determination of the Activities of SOD, CAT and POD

5.0 g of pepper tissue was homogenate with 10 mL extraction solution (PBS, pH 6.8, 1% PVP for POD; PBS, pH 7.8, 1% PVP for SOD and CAT), then centrifuged at 4000 rpm for 20 min. The absorbance values of the filtrates were measured at 240 nm, 470 nm and 560 nm to evaluate the activities of CAT, POD and SOD respectively, according to the method described by Taheri et al. [54]. The change of absorbance value of 0.1 per minute was considered as one POD enzyme activity unit (U). The catalytic decomposition of 1 µmol H_2_O_2_ per minute of the sample was defined as one CAT enzyme activity. 50% inhibition of NBT photoreduction was considered as one SOD enzyme activity. The activity of POD, SOD and CAT was expressed as 10^3^ U kg^–1^.

### 4.7. Determination of AsA-GSH Cycle

The contents of AsA, dehydroascorbate (DHA), reduced glutathione (GSH) and oxidized glutathione (GSSG), plus the activities of APX, GR, DHAR and MDHAR were assayed with their respective commercial biochemical kits (Suzhou Michy Biotechnology Co. Ltd., Suzhou, China), in accordance with the kit instructions. At 290 nm, oxidation of 1 μmol AsA per minute by the sample was defined as one APX enzyme activity. At 265 nm, the generation of 1 nmol AsA per minute of reduction was defined as one DHAR enzyme activity. At 340 nm, the reduction of 1 nmol GSSG to generate 2 nmol GSH per minute was defined as one GR enzyme activity. At 340 nm, oxidation of 1 nmol NADH per minute was defined as one MDHAR enzyme activity. The results were expressed as 10^3^ U kg^–1^.

### 4.8. Statistical Analysis

All experiments were repeated thrice and followed a completely randomized design. Data shown are the means ± standard error (SE) of the three biological replicates. Data were analyzed by one-way ANOVA and the significance of difference between the data sets analyzed at the level of * *p* < 0.05 or ** *p* < 0.01. SIMCA software (v17, Umetrics, America) was used to analyze projection to latent structures discriminant analysis (PLS-DA) and orthogonal projection to latent structures discriminant analysis (OPLS-DA) of ROS metabolism-related indicators. Heat maps were created using Tbtools software (v1.09876) created by Chengjie Chen (Guangzhou, China).

## 5. Conclusions

Our study showed that MT application was effective in reducing the occurrence of CI and preventing the loss of nutritional quality due to extensive accumulation of ROS and MDA in postharvest green horn pepper. These positive effects were attributed to MT-induced antioxidant systems, such as enhanced CAT, POD and SOD activities, as well as promotion of the AsA–GSH cycle activity. The results suggest that exogenous MT treatment can be used to reduce postharvest pepper loss by triggering the antioxidant system, thereby alleviating CI and maintaining fruit quality of green horn pepper during cold storage. We are also hopeful that MT receive permission for use in green horn peppers soon.

## Figures and Tables

**Figure 1 plants-11-02367-f001:**
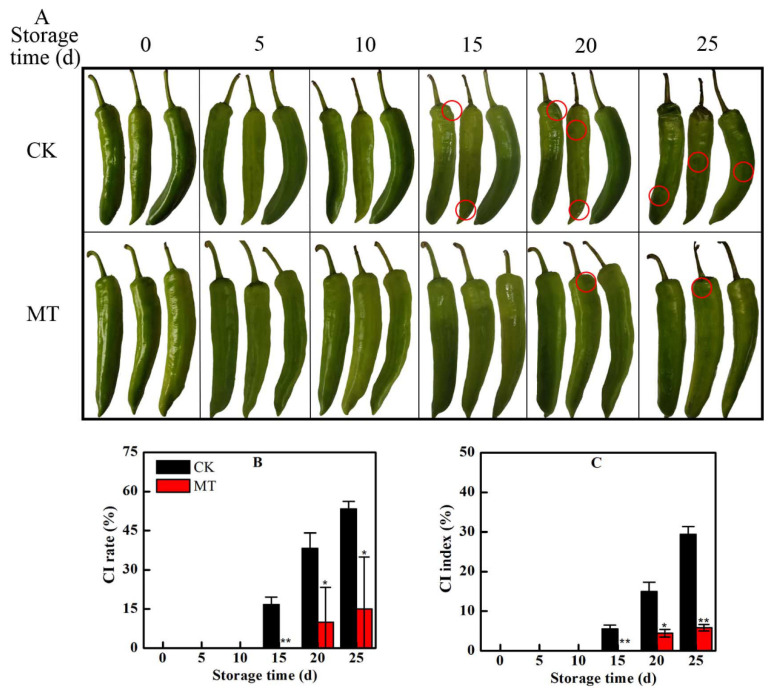
Change in the surface appearance due to chilling injury (CI) (**A**), CI rate (**B**), and CI index (**C**) of horn pepper fruit treated with water (CK) or melatonin (MT). Positions in which the CI symptoms appear on the surface of the pepper fruit are circled in red. Asterisks * and ** indicate the significant differences (*p* < 0.05 or *p* < 0.01, respectively) between treatments.

**Figure 2 plants-11-02367-f002:**
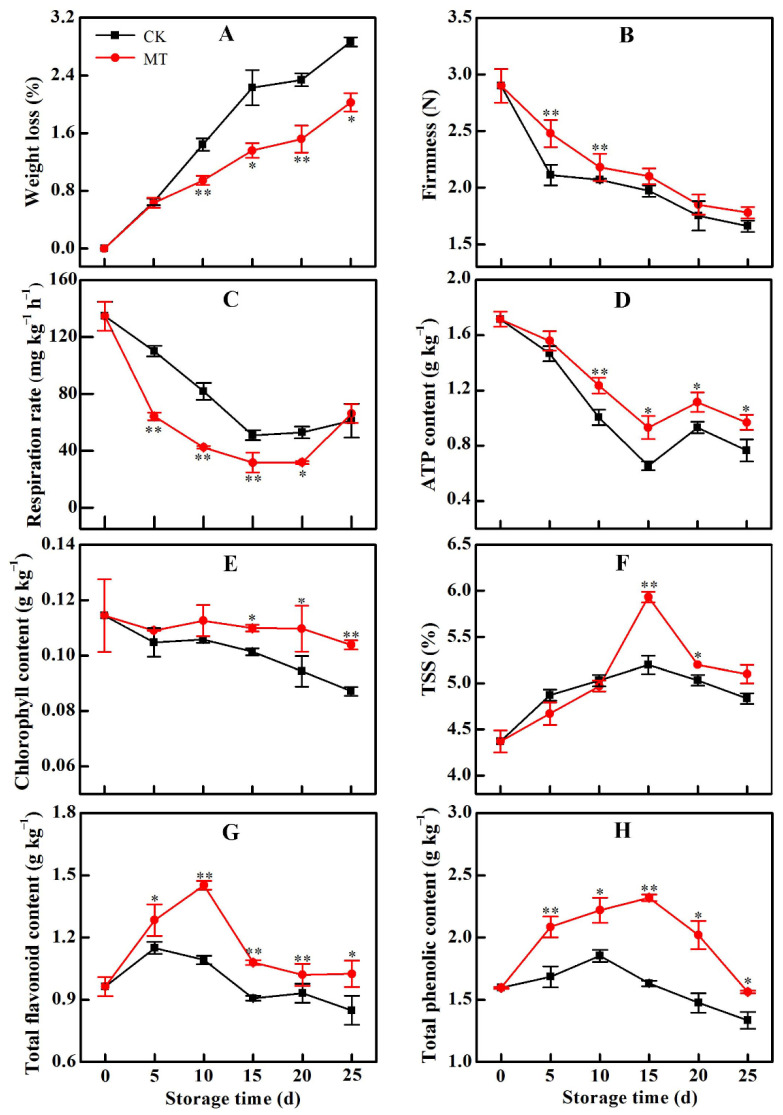
Change in weight loss rate (**A**), fruit firmness (**B**), tissue respiration rate (**C**), cellular ATP content (**D**), chlorophyll content (**E**), total soluble solids (TSS) content (**F**), flavonoid content (**G**) and total phenolic content (**H**) of pepper fruit during cold storage after treatment with water (CK) or melatonin (MT). Asterisks * and ** indicate the significant differences (*p* < 0.05 or *p* < 0.01, respectively) between treatments.

**Figure 3 plants-11-02367-f003:**
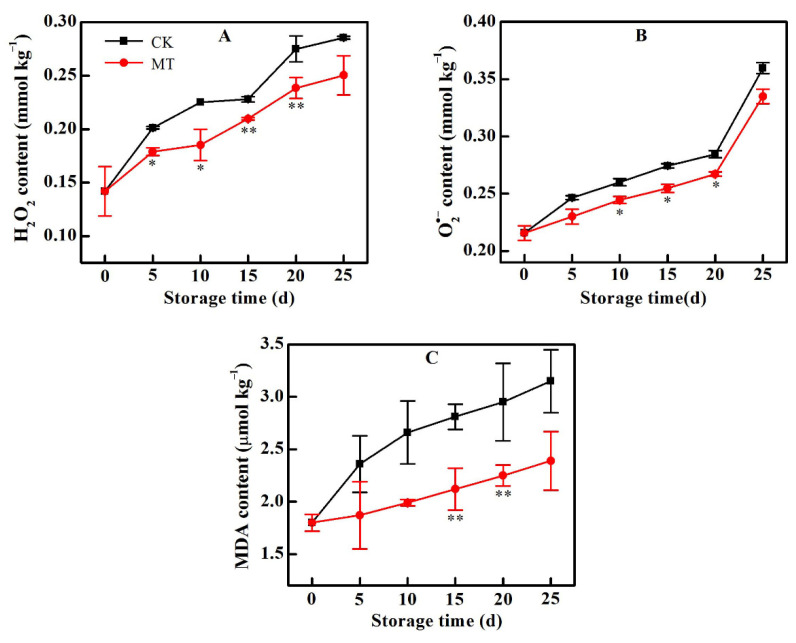
Change in the contents of H_2_O_2_ (**A**), O_2_^•^^−^ (**B**) and MDA (**C**) in horn pepper fruit during cold storage after treatment with water (CK) or melatonin (MT). Asterisks * and ** indicate the significant differences (*p* < 0.05 or *p* < 0.01, respectively) between treatments.

**Figure 4 plants-11-02367-f004:**
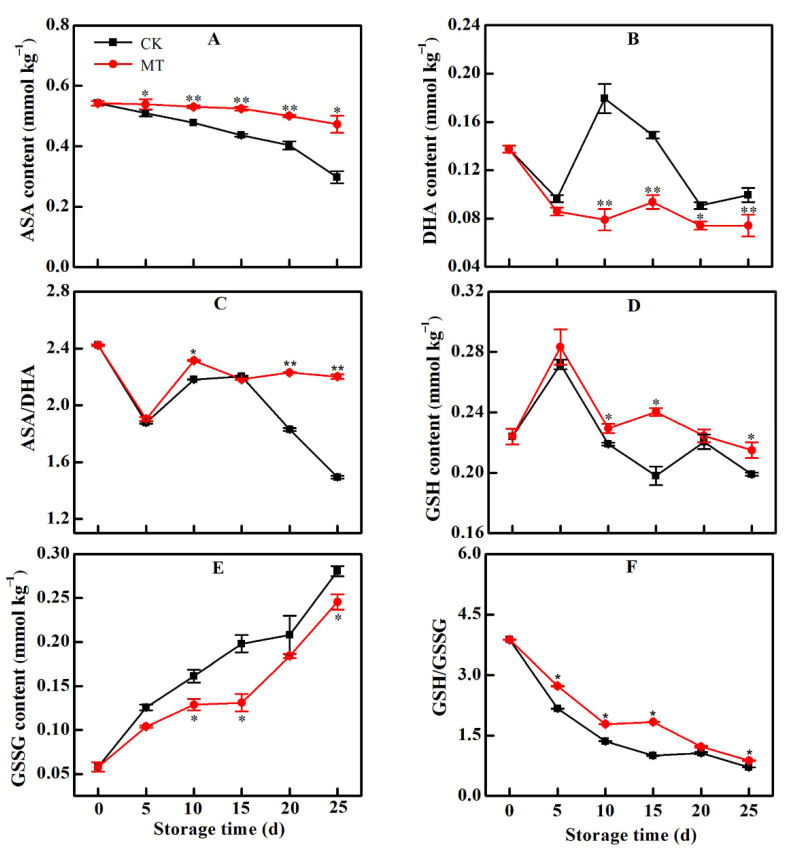
Change in the levels of AsA (**A**), DHA (**B**), AsA/DHA (**C**), GSH (**D**), GSSG (**E**) and GSH/GSSG (**F**) in horn pepper fruit during cold storage after treatment with water (CK) or melatonin (MT). Asterisks * and ** indicate the significant differences (*p* < 0.05 or *p* < 0.01, respectively) between treatments.

**Figure 5 plants-11-02367-f005:**
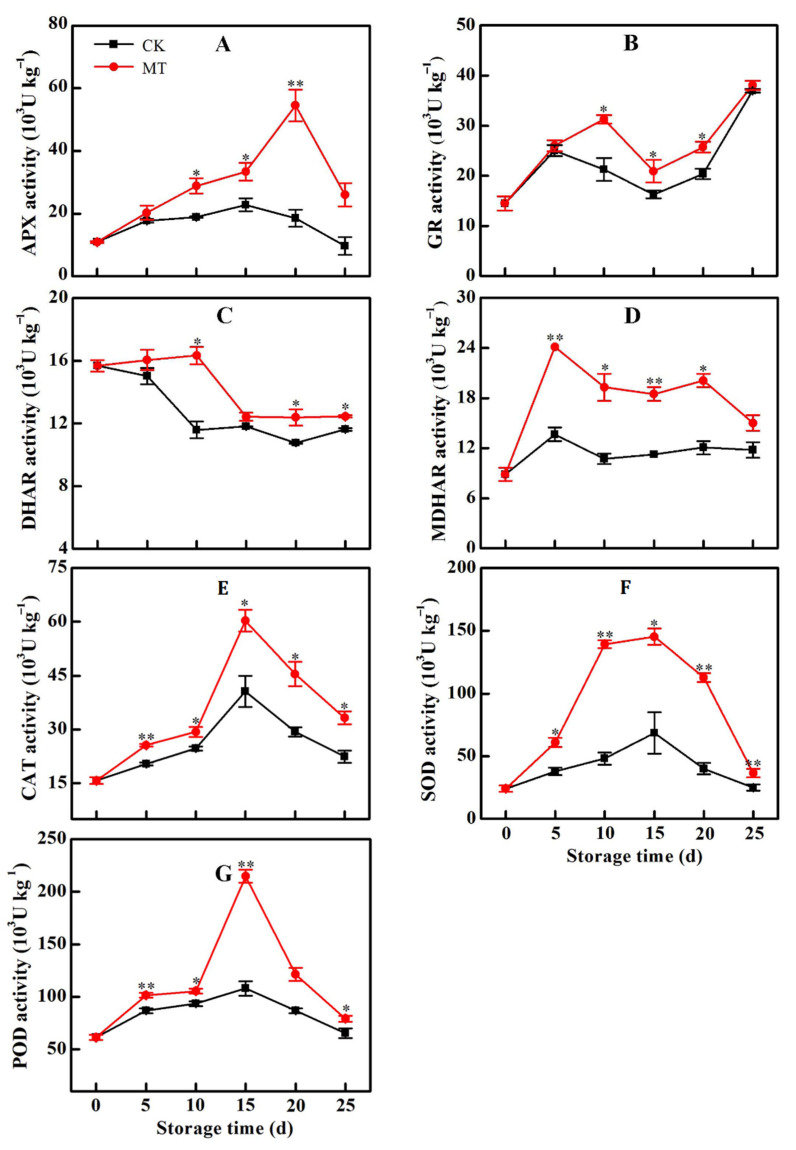
Change in the activities of APX (**A**), GR (**B**), DHAR (**C**), MDHAR (**D**), POD (**E**), CAT (**F**) and SOD (**G**) in horn pepper fruit during cold storage after treatment with water (CK) or melatonin (MT). Asterisks * and ** indicate the significant differences (*p* < 0.05 or *p* < 0.01, respectively) between treatments.

**Figure 6 plants-11-02367-f006:**
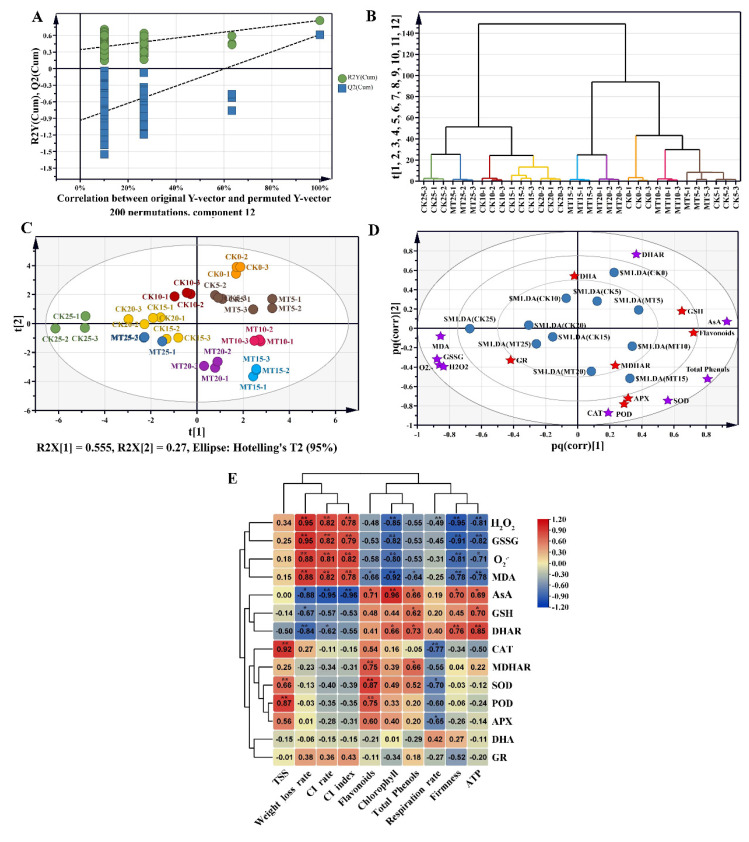
Correlation analysis of ROS metabolism with pepper CI and fruit quality. (**A**) permutations plot of PLS-DA, (**B**) hierarchical clustering analysis plot of PLS-DA, (**C**) score scatter plot of PLS-DA, (**D**) loading scatter plot of PLS-DA, (**E**) correlation analysis of ROS metabolism related indicators with pepper CI and fruit quality. The blue dots in (**D**) represents different samples, and the asterisks represent different ROS metabolism-related indicators, red being for VIP > 1 and purple for VIP < 1.

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
