# Peer review of "Exogenous Application of Melatonin to Green Horn Pepper Fruit Reduces Chilling Injury during Postharvest Cold Storage by Regulating Enzymatic Activities in the Antioxidant System"

_plants, 2022, doi:10.3390/plants11182367_

Round 1

Reviewer 1 Report

The manuscript studied the use of exogenous melatonin to reduce chilling injury and fruit quality loss in postharvest horn pepper by regulating the antioxidant system. Its topic is interesting and important. The manuscript is systematic and complete, and its results have practical implications for postharvest reduction of pepper losses. I have some comments that the authors should addressed.

1.        The content of the introduction section seems redundant. For example, lines 29-39 about horn pepper, lines 40-64 about ROS, and lines 65-79 about melatonin need to be compressed. There is no need to write so much, many concepts are known to the readers.

2.        It is suggested to add a paragraph describing the industrial application of melatonin for pepper preservation, including the significance and possible problems in its application.

3.        The definition of enzymatic activity needs to be provided.

4.        It is recommended to analyze the main causes of chilling injury in peppers based on the results of this study.

5.        How is energy metabolism involved in the occurrence of cpepper chilling injury?

6.        Other comments:

l  Line 21, ‘ascorbate–glutathione (AsA–GSH)’ change to ‘AsA–GSH’

l  Line 23, replace ‘ascorbate’ with ‘ascorbic acid’

l  Line 37, a space is required before °C

l  The reference part in Line 148, 314, 337, need to be revised.

l  Line 202, ‘Figure 6F’ change to ‘Figure 6E’

l  The Y-axis ‘O2.-’ in Figure 3B need to be modified

l  ‘ASA’ in Figure 4 should change to ‘AsA’

l  The expression of ‘O2•−’ in the manuscript needs to be unified

l  Line 481, the journal name needs to be italicized

l  Line 483, the format of author names in references 49 needs to be modified

Author Response

Dear  Reviewer:

Thank you for your letter and for the reviewers’ comments concerning our manuscript entitled “Exogenous Melatonin Reduces Chilling Injury and Fruit Quality Loss in Postharvest Horn Pepper by Regulating the Antioxidant System” (plants-1896675). We have studied the comments carefully and have made corrections according to the comments and suggestions which hope the manuscript is now suitable for Plants. All revised portions are marked in red in the paper. The main corrections in the paper and responses to the  reviewer’s comments are as follows:

  1. The content of the introduction section seems redundant. For example, lines 29-39 about horn pepper, lines 40-64 about ROS, and lines 65-79 about melatonin need to be compressed. There is no need to write so much, many concepts are known to the readers.

Answer: We have refined this part of these descriptions, and the revised portions are marked in red in the paper.

  1. It is suggested to add a paragraph describing the industrial application of melatonin for pepper preservation, including the significance and possible problems in its application.

Answer: Dear reviewer, as we know, MT has not been applied in the pepper industry at present, it is still in the stage of scientific research. So, we added the related description of the advantages of MT as a preservative in the industry and the significance of MT in reducing pepper loss. The limitations of MT in the industrial application may involve its half-life, light avoidance and dose accumulation effects, etc, these may be our later concerns and are not covered in this paper, so we did not add it in the introduction. We hope you will agree with our explanation!

  1. The definition of enzymatic activity needs to be provided.

Answer: We have supplemented all definitions of enzymatic activity in Methods.

  1. It is recommended to analyze the main causes of chilling injury in peppers based on the results of this study.

Answer: Based on our results, we conclude that ROS oxidative damage is the main cause of the occurrence of peppers CI. We made some analyses of the main causes of pepper CI in Discussion 3.2, and we further supplemented the analyses. Considering the effects of respiration, transpiration and reduced firmness on fruit storability in the early stage and the accumulation of ROS to a certain extent will cause cellular damage, while CI appears only in the middle and late storage period, we speculate that low temperature storage may firstly affect the postharvest metabolism of fruit, and this effect accelerated the accumulation of ROS, while ROS accumulation to a certain extent accelerated membrane lipid peroxidation damage, leading to the appearance of CI symptoms. The MT treatment induced activation of the antioxidant system, thus reducing the accumulation of ROS and CI further supporting our view.

  1. How is energy metabolism involved in the occurrence of pepper chilling injury?

Answer: In our study, the APT content decreased rapidly during the first 15 d of storage and then fluctuated but remained low. Interestingly, we found that pepper fruit also showed CI symptoms at 15 d. In contrast, MT treatment inhibited the decrease of ATP and maintained higher energy levels throughout the storage period. Increasing evidence has shown that higher levels of ATP and energy charge had a positive impact on the enhancement of chilling tolerance in different fruit like zucchini [1], banana [2], as well as nectarines [3]. ATP is the major energy source of biological organisms maintains cell metabolism because ATP acts the center of metabolism, storage and utilization of energy. Changes in cellular energy metabolism directly affect the formation of membrane lipids and the integrity of cell membranes. Long-term storage at low temperatures can reduce the ATP content of the fruit, resulting in disruption of energy production, transport and dissipation, and the disruption of various energy-consuming metabolisms. In contrast, maintaining high energy levels of fruit can reduce fruit CI. All these results indicate the close relationship between energy metabolism and fruit CI. Therefore, maintaining a higher energy level is beneficial to protect fruit from rapid quality loss due to low temperature stress. This agrees with our study, and the effect of energy on green pepper CI was also discussed in Discussion 3.1.

1 Zuo, X.; Cao, S.; Jia, W.; Zhao, Z.; Jin, P.; Zheng, Y. Near-saturated relative humidity alleviates chilling injury in zucchini fruit through its regulation of antioxidant response and energy metabolism. Food Chem. 2021, 351, 129336.

2 Liu, J.; Lia, F.; Lia, T.; Yun, Z.; Duana, X.; Jiang, Y. Fibroin treatment inhibits chilling injury of banana fruit via energy regulation. Sci. Hortic. 2019, 248, 8–13.

3 Zhang, W.; Zhao, H.; Jiang, H.; Xua, Y.; Cao, J.; Jiang, W. Multiple 1-MCP treatment more effectively alleviated postharvest nectarine chilling injury than conventional one-time 1-MCP treatment by regulating ROS and energy metabolism. Food Chem. 2020, 330, 127256.

  1. Other comments:

Line 21, ‘ascorbate–glutathione (AsA–GSH)’ change to ‘AsA–GSH’

Answer: Done as suggested.

Line 23, replace ‘ascorbate’ with ‘ascorbic acid’

Answer: Done as suggested.

Line 37, a space is required before °C

Answer: Based on comment 1 of reviewer 1, we have refined the description about horn pepper, in the revised introduction, we deleted ‘when they are exposed to low temperatures (< 7°C) for long periods of time’.

The reference part in Line 148, 314, 337, need to be revised.

Answer: I have revised all these reference mistakes in the paper.

Line 202, ‘Figure 6F’ change to ‘Figure 6E’

Answer: Done as suggested.

The Y-axis ‘O2.-’ in Figure 3B needs to be modified

Answer: I have changed the ‘O2.-’ in Figure 3B to ‘O2.-’.

‘ASA’ in Figure 4 should change to ‘AsA’

Answer: Done as suggested.

The expression of ‘O2•−’ in the manuscript needs to be unified

Answer: I have unified the expression of ‘O2•−’ in the paper.

Line 481, the journal name needs to be italicized

Answer: I have revised the references 47.

Line 483, the format of author names in references 49 needs to be modified

Answer: I have revised the references 48.

Reviewer 2 Report

The research hypothesis is quite novel. Authors did a lot of nice work. However, English writing is quite poor and is very hard to follow. It took me one hour to just finish one page. The manuscript needs to be improved a lot with the help of a native English speaker. It is not ready for our review yet. 

Author Response

Dear reviewer:

We are sorry that our poor writing made you unwilling to continue to review this manuscript. Thank you for revising the title and abstract of the manuscript. We have carefully revised and reviewed it with English writing peers. We hope that the current revised manuscript can meet the requirements for review and publication.

Responses to comments

  1. I think you missed a step here.

Answer: Thanks for pointing this out, we have added experimental treatment in the abstract.

  1. Symptoms should be things that you can see. I do not think you can see nutrient loss with your eyes. They are typically measured with an instrument.

Answer: We agree with you and have removed ‘and nutrient loss’ from the description.

  1. What do you mean by ‘Under normal circumstances’?

Answer: Sorry for the inappropriate expression, what we want to say is that plants themselves can relieve ROS stress through their antioxidant system under stress-free conditions. We have revised the expression.

We tried our best to improve the manuscript and made some changes in the manuscript, and marked them in the revised paper. We appreciate for Editors/Reviewers' warm work earnestly, and hope that the correction will meet with approval. Once again, thank you very much for your comments and suggestions.

Yours

Sincerely

Xiaoli Tan

Round 2

Reviewer 2 Report

The revised version looks much better. I made some changes to the manuscript. Please refer to the attached file for more information.

One big question mark is the likelihood of MT being approved for commercial use. Will the government likely grant its approval in the near future? Please address this issue in your discussion.

Author Response

Responses to the reviewer’s comments:

Reviewer

Dear reviewer:

Thank you very much for agreeing with our scientific research results and the revised manuscript, and for making changes to the manuscript. We have revised the full text according to your revision suggestion, all revised portions are marked in green based on the uploaded revised manuscript. We hope the current revised manuscript can meet the requirements for review and publication.

Yours

Sincerely

Xiaoli Tan